# Effect of Different Cooking Methods on the Bioactive Components, Color, Texture, Microstructure, and Volatiles of Shiitake Mushrooms

**DOI:** 10.3390/foods12132573

**Published:** 2023-06-30

**Authors:** Fen Yao, Hong Gao, Chao-Min Yin, De-Fang Shi, Xiu-Zhi Fan

**Affiliations:** 1Institute of Agro-Products Processing and Nuclear-Agricultural Technology, Hubei Academy of Agricultural Sciences, Wuhan 430064, China; yaofenyou@163.com (F.Y.); highong@163.com (H.G.); yinchaomin@163.com (C.-M.Y.); rongshusan@126.com (D.-F.S.); 2Shanxi Key Laboratory of Edible Fungi for Loess Plateau, Jinzhong 030801, China

**Keywords:** shiitake mushroom, bioactive components, color, texture, microstructure, volatiles

## Abstract

The effects of different cooking methods (steaming, boiling, air frying, and oven baking) and cooking times (0, 5, 10, 15, and 20 min) on the bioactive components (total phenol, total flavonoid, crude polysaccharides, and eritadenine), color, texture, microstructure, and volatiles in shiitake mushrooms were investigated in this study. Steaming, boiling, and air frying for 5–20 min could decrease the contents of all the four bioactive components in the shiitake mushroom. However, oven baking for 5 min and 10 min showed the highest contents of total phenolics and total flavonoids, respectively. Moreover, the lowest losses of crude polysaccharides and eritadenine were observed for oven baking for 5 min and 15 min, respectively. The lightness of shiitake mushrooms was decreased by all treatments; however, steaming could keep a higher brightness compared with other methods. The microstructure was damaged by all cooking methods, especially air frying for 20 min. Meanwhile, steaming for 20 min decreased the hardness mostly, and there was no significant difference with air frying for 20 min. All cooking treatments decreased the complexity of the flavors and the relative contents of volatile compounds; the lowest contents were found when boiling for 5 min. From these results it can be seen that the physical, histological, and chemical features in shiitake mushroom were influenced by cooking methods and times. In addition, our results provide valuable information for the cooking and processing of shiitake mushrooms and other fungi.

## 1. Introduction

The shiitake mushroom (*Lentinula edodes*) is one of the most important commercially cultivated edible mushrooms worldwide [1]. It is rich in nutrients, such as dietary fiber, protein, amino acids, mineral elements, vitamins, etc. [2]. It also provides several physiological functions, such as enhancing body immunity, anti-tumor, anti-oxidation, anti-inflammatory, and anti-virus properties, and so on [3]), these functions were proved to be related to the bioactive components in shiitake mushroom. Among these bioactive components, phenolic compounds were found to exhibit potential activities, including anti-inflammatory, antioxidant, anticarcinogenic, and antihemolytic functions [4]. Polysaccharides including β-glucans, chitins, etc. from shiitake mushroom also demonstrated various pharmacological features such as anti-cancer, anti-neurodegenerative, antiviral, and hypocholesterolemic functions [3]. In addition, eritadenine, a purine alkaloid first isolated from the shiitake mushroom, was proved to have hypocholesterolemic effects and could be used to prevent or treat hypertension in humans [5].

As an edible fungus, shiitake is usually eaten after cooking rather than raw. Moreover, it is more delicious and digestible after cooking. Nowadays, mushrooms are usually cooked by heat treatments such as boiling, steaming, stir frying, and others. Studies have shown that the cooking temperature and time were crucial for the edible quality of food, and different cooking conditions could affect the extent of chemical reactions and the physical properties [6]. Furthermore, the cooking process can damage the active constituents or increase the content of bioactive compounds [7,8]. Manzi et al. [9] and Choi et al. [10] found the polyphenolic compounds in mushrooms were lost and the antioxidant capacities were increased after boiling, while polyphenol concentrations in shiitake mushrooms and the antioxidant properties were increased after heat treatment. Moreover, it has been proved that culinary treatments including steaming, frying, baking, and grilling could change not only the physicochemical properties but also the histological properties and flavor profile of mushrooms [11,12]. However, according to the current information, it is hard to reach a single conclusion on the advantages and disadvantages of different cooking methods and conditions on the food quality of shiitake mushrooms. 

Therefore, to evaluate the effects of heat treatment on the physical, histological, and chemical features of shiitake mushrooms, four types of culinary methods, including steaming, boiling, air frying, and oven baking, were used to process the shiitake mushrooms, and the cooking times were set as 5, 10, 15, and 20 min. After cooking, the color, texture, volatiles, and microstructure of the shiitake mushrooms were analyzed, and the contents of the bioactive components (phenolic compounds, polysaccharides, and eritadenine) were detected. We hope to provide theoretical guidance for the scientific processing and the appropriate cooking of shiitake mushrooms and other edible fungi.

## 2. Materials and Methods

### 2.1. Materials, Cooking Methods, and Sample Preparation

Shiitake mushrooms were cleaned by sterile water and cut into 3 mm pieces. A quantity of 100 g of raw shiitake mushrooms were used as the control group, and 100 g of raw shiitake mushrooms cooked with different methods were set as the treatment groups, namely, group I, steamed for 5, 10, 15, and 20 min at 100 °C, respectively; group II, boiled for 5, 10, 15, and 20 min at 100 °C, respectively; group III, air-fried for 5, 10, 15, and 20 min at 100 °C, respectively; and group IV, oven-baked for 5, 10, 15, and 20 min at 100 °C, respectively. After cooking, half of the samples were freeze-dried and ground into powders for bioactive compounds analysis (total phenol, total flavonoid, polysaccharides, and eritadenine), and the rest were stored at 4 °C for color determination, SEM observation, texture, and GC-MS analysis. Meanwhile, the control group samples were also lyophilized and stored at 4 °C for use.

### 2.2. Analysis of Bioactive Components in Shiitake Mushrooms

#### 2.2.1. Preparation and Determination of Total Phenol Content 

The total phenol content (TPC) of each sample was extracted as follows. Briefly, 0.1 g dried mushroom powders were mixed with 3 mL of a solution consisting of ethanol and deionized water (70:30, *v*/*v*), and the mixture was sonicated at 300 W for 10 min at room temperature. The supernatant was collected after centrifugation for 25 min at 4 °C, 12,000× *g*. The extraction procedures were all repeated three times, and the filtrates were merged. The TPC of each extract was quantified using the Folin–Ciocalteu method described by Nithiyanantham et al. with slight modifications [13]. Briefly, 125 μL of Folin–Ciocalteu reagent (0.2 M) was mixed with 25 μL of extract or gallic acid standard (25–250 μg/mL). After 10 min, 12 μL of 10% Na_2_CO_3_ was gently added, and the mixture was incubated for 2 h before the absorbance was recorded at 765 nm on a UV-visible (UV-Vis) plate reader (Spark, Switzerland). TPC was expressed as milligrams of gallic acid equivalents (GAE) per gram of dry weight of mushroom powders (mg GAE/g DW).

#### 2.2.2. Preparation and Determination of Total Flavonoid Content

The total flavonoid content (TFC) of each sample was extracted as mentioned above (Section 2.2.1). The TFC of each extract was evaluated using the aluminum chloride colorimetry method as described by Tai et al. with slight modifications [14]. In brief, 110 μL of NaNO_2_ (0.066 M) was mixed with 25 μL of extract or rutin standard (200–800 μg/mL), and the mixture was incubated for 5 min at room temperature. Then 15 μL of AlNO_3_ (0.75 M) was added and incubated for 6 min. The reaction was stopped by the addition of 100 μL of 0.5 M NaOH solution. The absorbance was recorded at 510 nm on a UV-Vis plate reader. TFC was expressed as milligram of rutin equivalents (RE) per gram of dry weight of mushroom powders (mg RE/g DW).

#### 2.2.3. Preparation and Determination Crude Polysaccharides Content

Dried shiitake mushroom powders (1 g) were extracted with 20 mL ethanol by using an ultrasound machine, with a power of 300 W at room temperature for 30 min. Then, the extract solution was centrifuged at 4000× *g* for 10 min. Subsequently, the precipitate was extracted with 50 mL distilled water at 100 °C for 2 h. The precipitate was washed twice with distilled water. After centrifugation, the supernatant was collected and added to 100 mL deionized water. The crude polysaccharides content was determined by the phenol-sulfuric acid method described by Nielsen [15].

#### 2.2.4. Preparation and HPLC Analysis of Eritadenine

The extraction and quantification of eritadenine in the shiitake mushroom samples were carried out as mentioned by Enman et al. [5]. The dried mushroom powders were soaked in 5% ethanol for 4 h (*w*/*v*, 1:100) and then extracted using ultrasonic-assisted technology (500 W) for 40 min at room temperature. The extracts were centrifuged at 12,000 g for 20 min. The filtrates then underwent HPLC analysis for eritadenine quantification after filtering through a 0.22 µM membrane filter. 

The concentration of eritadenine was assessed by a Shimadzu LC-20AT HPLC system (Shimadzu, Japan) equipped with a quaternary pump and Inertsil ODS-SP C18 column (5 µm, 4.6 mm × 250 mm) (Shimadzu, Japan). The mobile phase was composed of 8 mM phosphate buffer solution (pH 4.5) and methanol (*v*/*v*, 95:5). The column was operated at 30 °C at a constant flow rate of 0.8 mL/min. The injection volume was 10 µL, and the effluents were monitored at 260 nm.

### 2.3. Color Determination of Shiitake Mushroom Samples

The color parameters of the raw and cooked samples treated with different cooking methods and times (Section 2.1, stored at 4 °C) were measured using Chroma Meter CR-400 (Konica Minolta, Osaka, Japan) based on the previously described method [16]. The colorimeter was calibrated by using a standard white before the color parameters were recorded, and fifteen replicate measurements were performed. Three parameters (L*, a*, b*) were utilized to present the color differences in distinct shiitake mushroom samples.

### 2.4. Texture Analysis of Shiitake Mushroom Samples

The textures of the raw and cooked shiitake mushrooms were tested according to the method by Lespinard et al. [17]. The textures (Section 2.1, stored at 4 °C) were tested at room temperature (TA-XT Plus, Texture Technologies, Hamilton, MA, USA), and 10 replicates were conducted for each group. The test conditions were as follows: the P/36R probe had a mode of TPA, a pretest speed of 2.0 mm/s, a test speed of 1.0 mm/s, and a post-test speed of 1.0 mm/s; the shape variable was set as 70%; and the trigger force was 0.5 N. The detected parameters were hardness, springiness, chewiness, resilience, and gumminess. 

### 2.5. SEM Observation of Shiitake Mushroom Samples

The raw and treated samples were cut into 3 × 3 × 2 mm pieces and fixed with 2.5% glutaraldehyde solution at 4 °C for 24 h. The fixed samples were rinsed with 0.1 M PBS (pH 7.4) three times and then dehydrated in ethanol with an increasing concentration gradient of 30%, 50%, 70%, 80%, 90%, 95%, and 100% for 15 min and finally in isoamyl acetate for 15 min. Moreover, the samples were dried with a critical point dryer. The dried samples were coated with gold for 30 s. The cross section of the sample was observed with a scanning electron microscope (HITACHI, SU8100, Hitachi High-Tech, Hitachi, Japan).

### 2.6. GC-MS Analysis

HS-SPME-GC-MS was used for the determination of volatile compounds in shiitake mushrooms based on the procedure described by Yin et al. with some modification [18]. The raw shiitake mushrooms and cooked shiitake mushrooms (5 g) were homogenized in saturated sodium chloride (20 mL). All samples were balanced at 60 °C for 10 min, and then the fiber was inserted to the vital to extract volatile compounds for 40 min. Subsequently, the fiber was inserted into the injection port of GC and desorbed for 5 min under the splitless mode.

The chromatography conditions were as follows: The separation and detection of volatile compounds were carried out by using DB-5 MS chromatographic columns (60 m × 0.25 mm × 0.25 μm; Agilent Technologies, Santa Clara, CA, USA) by gas chromatography tandem mass spectrometry (Agilent 7890A- 5973C, GC-QQQ-MS). No split mode was used for each injection. Helium (99.99%) was used as the carrier gas (1 mL/min). The initial temperature of the oven was kept at 40 °C for 2 min, rose to 180 °C at a rate of 3 °C/min, was kept for 2 min, and increased to 240 °C at the rate of 10 °C/min. The mass spectrometry conditions were as follows: detector interface temperature, 230 °C; ion source temperature, 150 °C; ionization energy, 70 eV; electron multiplier voltage, 350 V; and filament emission electric current, 200 μA. By comparing with the compounds in National Institute of Standards and Technology 08. L (NIST08. L) mass spectrum library, the mass spectra of volatile components in the samples were qualitatively identified. The acceptance criteria for matching scores were higher than 80%. 

### 2.7. Statistical Analysis

Three replicates were applied in all the experiments, except for the color and texture assays. The data were expressed as the mean ± SD, and the significances of the differences were analyzed using one-way analysis of variance and Duncan’s multiple-range tests. *p* values of <0.05 were considered statistically significant.

## 3. Results and Discussion

### 3.1. Effects of Different Cooking Methods on Bioactive Component Contents

The contents of total phenol, total flavonoid, crude polysaccharides, and eritadenine in raw and treated shiitake mushrooms under different cooking conditions were analyzed (Figure 1). In the cooked shiitake mushrooms, the TPC ranged from 92.81 to 97.02, 55.17 to 79.57, 93.23 to 142.65, and 152.53 to 222.13 mg GAE/g DW after steaming, boiling, air frying, and oven baking, respectively. Compared with the raw sample (127.08 mg GAE/g DW), oven-baked shiitake mushrooms for 5 min showed the highest TPC (222.13 mg/g DW) with an increase of 75%. In the boiled shiitake mushrooms, the TPC had the most loss at 15 min, and the reduction rate was 57%. Meanwhile, steaming for 5–20 min and air frying except for 10 min also showed a reduction in TPC, and the maximum loss rate was 56% and 27%, respectively. These results were similar to the report that steaming, boiling and air frying reduced the TPC in broccoli [19]. The most loss of TPC in boiled shiitake mushrooms may be due to leaching of the soluble phenols into boiling water. Wachtel-Galor et al. also reported that boiling reduced the TPC of cabbage, broccoli, and Chinese cabbage by more than 60% [20]. On the other hand, the increase in the TPC during oven baking may be due to the destruction of cell membranes and walls, which leads to the increase in the release of phenols [21].

After steaming, boiling, air frying, and oven baking for 5–20 min, the TFC in cooked shiitake mushrooms ranged from 1.92 to 2.20, 1.45 to 1.69, 1.75 to 2.48, and 3.16 to 4.49 mg RE/g DW, respectively. In addition to steaming, all cooking processes caused the TFC losses to increase with the processing time. The raw shiitake mushroom presented the highest concentration of TFC (4.50 mg RE/g DW), which had no significant difference with the shiitake mushrooms baked in the oven for 5 min (4.49 mg RE/g DW). Similarly, Mashiane et al. also reported a reduction of the TFC in African pumpkin and pumpkin leaves after stir frying, boiling, steaming, and microwaving treatment [22]. All cooking methods had a negative effect on the TFC; this could be attributed to the degradation of compounds that were sensitive to heat [23].

Regarding crude polysaccharides, the contents in cooked shiitake mushrooms ranged from 2.93 to 3.81, 2.28 to 3.74, 4.83 to 5.13, and 4.67 to 5.57 g/100 g DW after steaming, boiling, air frying, and oven baking for 5–20 min, respectively. The raw shiitake mushrooms presented the highest content at 6.74 g/100 g DW. All cooking methods decreased the crude polysaccharides content, and the boiled mushrooms exhibited the most significant reduction (*p* < 0.05). The lowest concentration (2.28 g/100 g DW) was present in the mushrooms boiled for 20 min, resulting in a 66% reduction compared with the raw mushrooms. Lee et al. reported that the cell structure of shiitake mushrooms changed during boiling; therefore, it can be speculated that the content loss of polysaccharides was due to the release of crude polysaccharides stored in the cellulose network from the boiling tissue into the boiling water [24]. In addition, the minor loss (17%) of crude polysaccharides was observed after oven baking for 5 min; this may be associated with the oven causing the mushroom surface water to be quickly lost, cell contraction, and then limiting the loss of polysaccharides.

The concentrations of eritadenine in all cooked shiitake mushroom ranged from 298.39 to 359.03, 184.56 to 416.99, 406.04 to 523.24, and 476.88 to 551.31 mg/100 g DW after steaming, boiling, air frying, and oven baking for 5–20 min, respectively. The raw shiitake mushrooms had the highest eritadenine content (631.23 mg/100 g DW). All the cooked shiitake mushrooms showed a decrease in eritadenine when compared with the raw shiitake mushrooms. The minor loss of eritadenine was observed after oven baking for 15 min (551.31 mg/100 g DW), while the maximum loss was observed after boiling for 15 min (184.56 mg/100 g DW). Sori et al. [25] and Sánchez-Minutti et al. [26] reported that the eritadenine concentration of raw shiitake mushrooms decreased after frying, steaming, roasting, and boiling. Morales et al. [27] also found that the eritadenine content was reduced when boiling, compared with grilling. Eritadenine, a water-soluble molecule, can lixiviate into the aqueous medium. Therefore, the loss of eritadenine in all cooked shiitake mushrooms might be due to the lixiviation, degradation, or chemical transformation of the eritadenine under high temperature. 

### 3.2. Color Properties of the Shiitake Mushroom

Color is a key food quality parameter, which affects consumer preferences and the market value of the final products. The color data (L*, a*, b*) of the raw and all cooked shiitake mushrooms are shown in the Table 1. As shown in Table 1, the cooking method and time had significant effects on the color parameters. The raw shiitake mushroom was found to be brighter (L*: 84.21) than all cooked shiitake mushrooms with the value of L* ranging from 61.11 to 77.78. Steamed shiitake mushrooms showed relatively higher L* values than boiled, air-fried, and oven-baked mushrooms under the same cooking time. The air-fried and oven-baked mushrooms showed lower L* values, which may have been due to the production of brown pigments in the Maillard reaction during cooking [28]. Moreover, the L* values of the boiled shiitake mushrooms were decreased; this may likely have been because the high water content in the shiitake mushrooms might have caused the light to penetrate deeper into the tissues, resulting in a darker shiitake mushrooms surface [29].

Differently, the values of a* and b* in all cooked shiitake mushrooms were increased compared with those of the raw shiitake mushrooms, indicating the browning reactions. The a* and b* values of the shiitake mushrooms obtained by air frying for 20 min were the highest, which turned out to be 93% and 57% higher than those of the raw shiitake mushrooms, respectively. Jiménez-Zamora et al. [28] reported that the high temperatures and the loss of water in the air-frying and oven-baking treatments helped to achieve the activation energy required by Maillard and formed a yellowish-reddish compound. Moreover, the carotenoids concentration increased with the losing of water in mushrooms which could cause the color change [30]. The loss of pigment and the inactivation of polyphenol oxidase in the steamed and boiled shiitake mushrooms could also contribute to the increases in the values of a* and b* [17].

### 3.3. Texture 

Table 2 shows the effects of the cooking method and time on the textures of the cooked and raw shiitake mushrooms. The hardness of the raw shiitake mushrooms was 2007.33 g (force). Under all cooking methods, the hardness losses increased accordingly with the processing time. Compared with the raw shiitake mushrooms, the hardness was reduced by 21–63%, 13–49%, 40–63%, and 10–50% after steaming, boiling, air frying, and oven baking for 5–20 min, respectively. Similarly, the gumminess and chewiness values were reduced significantly. The decreased rates of gumminess in the shiitake mushrooms were 30–65%, 19–55%, 39–69%, and 5–54% after steaming, boiling, air frying, and oven baking for 5–20 min, respectively. The chewiness losses were 42–68%, 20–58%, 37–72%, and 5–63% for steaming, boiling, air frying, and oven baking, respectively. All cooking methods reduced the values of the hardness, gumminess, and chewiness, and the boiling treatment had relatively higher values of hardness, gumminess, and chewiness. Some studies have shown that the reductions of the hardness, gumminess, and chewiness in mushrooms and other plant products during heat treatment were due to the damage to the cell wall and tissue softening, which was weakened by protein denaturation, pectin dissolution, polysaccharide degradation, and other complex reactions [31]. Meanwhile, air frying exhibited the most damage to the hardness, gumminess, and chewiness of the shiitake mushroom compared with the other cooking methods, which may be attributed to shiitake mushrooms’ exposure to hot air leading the shrinkage of the mycelium.

The adhesiveness values in the cooked shiitake mushrooms increased, and the rates under steaming, boiling, air frying, and oven baking were 42–68%, 20–58%, 37–72%, and 5–63%, respectively, compared with the raw shiitake mushrooms. According to the reports of Nketia et al. [32], the increased adhesiveness might be related to the cell walls rupturing during cooking and the release of the polysaccharides with obvious viscosity characteristics, such as β-glucans and chitin.

The cohesiveness and springiness of the cooked shiitake mushrooms decreased gradually along with the time compared with the raw shiitake mushrooms. This result might be related to the collapse of cells and intercellular space, which restricted the bonds of proteins and polysaccharides [33].

In general, except for adhesiveness, the hardness, cohesiveness, gumminess, springiness, and chewiness of the cooked shiitake mushrooms decreased with the increasing cooking time. Moreover, compared with other cooking treatments, oven baking for 5 min had a relatively small loss in texture. 

### 3.4. SEM Observation of Shiitake Mushrooms

The microstructures of the shiitake mushrooms under different cooking methods and times were shown in Figure 2. The raw shiitake mushrooms were composed of round and regular mycelia, while all cooked shiitake mushrooms showed cellular tissue shrinkage. The flattened appearance of the shiitake mushroom mycelia during cooking could mainly be attributed to high cooking temperatures and the irreversible loss of cell membrane integrity due to dehydration [34]. Steaming, boiling, air frying, and oven baking had little effect on the microstructure of the shiitake mushrooms after cooking for 5 min, which possessed evenly distributed pores and less tissue collapse. However, more and more collapse and deformation of hyphae walls were observed when the shiitake mushrooms were cooked by steaming, boiling, air frying, and oven frying for 20 min. Similarly, Paciculli et al. [35] indicated that steaming caused cell separation and collapse of carrots when treated at 100 °C. Probably because of the evaporation of water from steaming, the porous structure that was formed caused capillary shrinkage stress. Meanwhile, the effects of boiling and oven baking on the structure found in this study were consistent with those observed in vegetables [36]. The structural damage induced by air frying was greater than that of other cooked shiitake mushrooms. Moreover, with the increase in the air frying time, the cellular tissue became more shrunken with large pores; this may have been due to the quick decrease in the water in the surface of the shiitake mushrooms, and severe shrinkage of shiitake mushroom occurred. Here the texture differences among different cooked mushrooms found by TPA analysis could be explained by the changes in the SEM observation. When cooking, the hardness decreased along with the damage to the mycelia structure. The highest hardness loss was present when the shiitake mushrooms were air fried for 20 min and the microstructure was most seriously damaged. 

### 3.5. HS-SPME-GC-MS Analysis of Volatile Compounds of Shiitake Mushrooms

As shown in Table 3, the volatile compounds, including 15 alcohols, 13 aldehydes, nine alkanes, four alkenes, six ketones, five sulfur compounds, three benzene compounds, and five others, were tentatively identified from the raw and cooked shiitake mushrooms. Among these compounds, 1-octen-3-ol, octanal, and oxime-, methoxy-phenyl- were common constituents in the raw and cooked shiitake mushrooms. The cooking method and time showed different effects on the volatile compounds in the shiitake mushrooms. As shown in Table 3, 30 compounds were detected in the raw shiitake mushrooms, mainly including alcohols, ketones, sulfur, aldehydes, and benzene compounds with the relative contents 44.89%, 15.74%, 5.43%, 2.82%, and 2.14%, respectively. Moreover, 1-octen-3-ol was the most prevalent volatile compound in the raw shiitake mushrooms, consistent with the report of Politowicz et al. [37], and the relative content was 41.33%. Cheng et al. [38] also reported that 1-octen-3-ol was the basic aroma compound in shiitake mushrooms, which was formed through the enzymatic degradation of linoleic acid, and it gave shiitake mushrooms an “earthy” and “mushroom-like” odor. 

Tian et al. [39] reported that boiling and oven baking led to a reduction in the volatile compounds compared with the raw shiitake mushrooms. In this study, similar results were obtained. After cooking for 5 to 20 min with the four methods, the variety of the volatile flavor substances was also decreased when compared with the raw shiitake mushroom. Specifically, 11–20, 9–18, 11–21, and 10–18 types of compounds were detected in steamed, boiled, air-fried, and oven-baked shiitake mushrooms, respectively. Furthermore, the volatile compounds in all the cooked mushrooms decreased mostly after cooking for 20 min. This may have been due to the degradation of the flavor component during heating treatment.

In addition, both the main volatile components and the relative contents were varied under different cooking methods and times. Benzenes were the major volatile compounds in the steamed and oven-baked mushrooms, with relative contents of 10.53–14.51% and 12.63–26.42% after 5 to 20 min of cooking, respectively. In the boiled mushrooms, alcohols were the main volatile compounds, and the relative contents were 23.06% after boiling for 10 min; 1-octen-3-ol was the major volatile compound with a relative content of 15.32% in the 15 min boiled mushrooms. In the air-fried mushrooms, the main volatile compounds were sulfur compounds (32.56%) after cooking for 5 min, and dimethyl trisulfide was the most prevalent volatile compound, with a relative content of about 27.72%. In sum, after cooking, the total volatile components in the mushrooms and the relative contents decreased. The least changes of flavor substances variety and relative content were present in the 5 min air-fried mushrooms. 

## 4. Conclusions

Shiitake mushrooms were steamed, boiled, air fried, and oven baked for 5–20 min, and the effects of these cooking treatments on the bioactive components, color, texture, microstructure, and volatiles were investigated. Among these cooking methods, only oven baking could improve the TPC and TFC and resulted in the least loss of crude polysaccharides and eritadenine contents. The color parameters showed the mushrooms darkened after cooking, especially boiling. In addition, all four cooking methods could cause texture damage and microstructure change; among them, air frying had a relatively larger impact. With respect to volatiles, all four cooking treatments decreased the variety of volatile components and their relative contents. 

In conclusion, oven baking was considered to be the best cooking method to improve or retain bioactive compounds. Steaming and air frying were the most promising methods for maintaining the color and volatiles, respectively. In addition, boiling had a relatively smaller negative influence on microstructure and hardness, compared with the other cooking methods. Each method had its own advantages. Therefore, the appropriate processing method can be selected according to the above research results and the processing purposes. 

In addition, although it was found that the cooking method could affect the bioactive component content in the shiitake mushrooms, the effect on their activity has not been determined. Therefore, it is necessary to conduct further studies to investigate the influence of cooking methods on the biological activities and bioaccessibility of active compounds, and then lay a theoretical basis and provide a foundation for the research and development of shiitake mushrooms.

## Figures and Tables

**Figure 1 foods-12-02573-f001:**
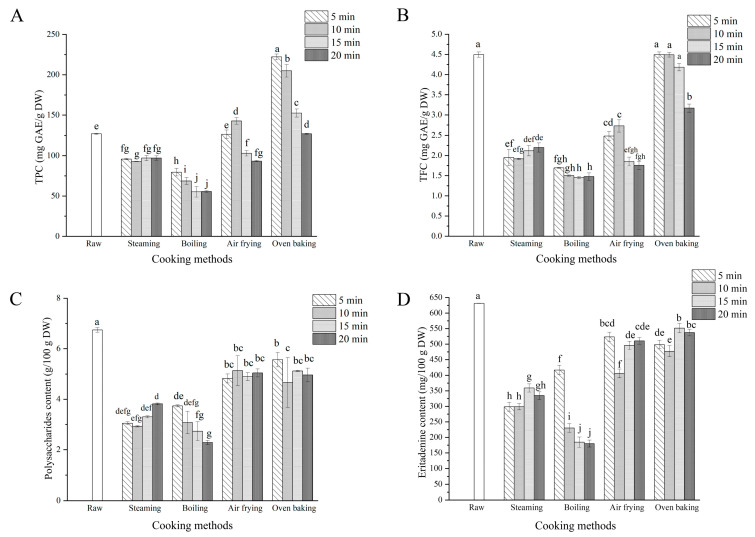
Content of total phenol (**A**), total flavonoid (**B**), crude polysaccharides (**C**), and eritadenine (**D**) in raw and cooked shiitake mushrooms. Lowercase letters above the columns represent the significant differences at *p* < 0.05.

**Figure 2 foods-12-02573-f002:**
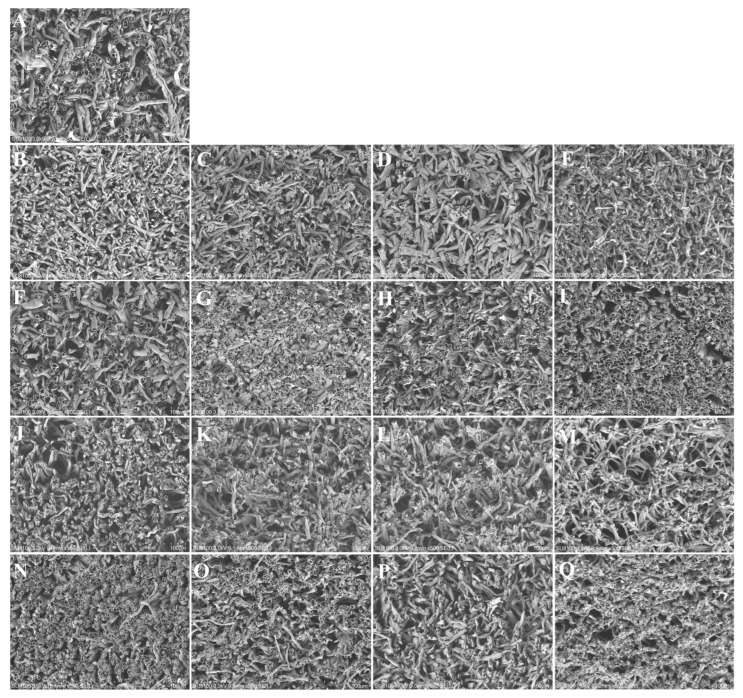
Effects of different cooking methods and times on the microstructure of raw and cooked shiitake mushrooms. (**A**) Raw shiitake mushroom; (**B**–**E**) steamed for 5, 10, 15, and 20 min, respectively; (**F**–**I**) boiled for 5, 10, 15, and 20 min, respectively; (**J**–**M**) air fried for 5, 10, 15, and 20 min, respectively; and (**N**–**Q**) oven baked for 5, 10, 15, and 20 min, respectively. Micrographs were taken at 500× magnification.

**Table 1 foods-12-02573-t001:** Colorimetric parameters of raw and cooked shiitake mushrooms.

Cooking Methods	Cooking Time (min)	L*	a*	b*
Raw	0	84.21 ± 2.92 ^a^	0.12 ± 0.08 ^ef^	6.34 ± 0.90 ^g^
Steaming	5	72.96 ± 4.62 ^de^	0.20 ± 0.03 ^ef^	12.3 ± 1.09 ^bc^
	10	75.84 ± 1.82 ^bcd^	1.12 ± 0.13 ^abcd^	14.16 ± 0.94 ^a^
	15	77.63 ± 2.06 ^bcd^	1.24 ± 0.26 ^abc^	13.23 ± 1.31 ^ab^
	20	77.78 ± 3.61 ^bcde^	1.35 ± 0.34 ^ab^	12.95 ± 0.75 ^ab^
Boiling	5	61.11 ± 8.05 ^e^	0.81 ± 0.02 ^g^	6.55 ± 1.65 ^g^
	10	71.59 ± 2.25 ^de^	0.25 ± 0.03 ^ef^	9.86 ± 0.86 ^ef^
	15	64.97 ± 4.71 ^fe^	0.22 ± 0.08 ^ef^	10 ± 1.80 ^def^
	20	67.00 ± 4.35 ^fe^	1.35 ± 0.09 ^ab^	11.97 ± 2.23 ^bc^
Air frying	5	72.05 ± 4.89 ^cde^	0.10 ± 0.03 ^f^	9.65 ± 1.89 ^f^
	10	75.79 ± 3.18 ^bcd^	0.44 ± 0.03 ^def^	10.52 ± 0.85 ^cdef^
	15	75.38 ± 2.18 ^bc^	1.17 ± 0.10 ^abcd^	13.07 ± 0.48 ^ab^
	20	72.53 ± 4.19 b^cde^	1.71 ± 0.05 ^a^	14.675 ± 0.67 ^a^
Oven baking	5	71.24 ± 4.50 ^de^	0.93 ± 0.06 ^bcde^	11.77 ± 1.60 ^bcd^
	10	75.23 ± 2.45 ^bcd^	0.53 ± 0.05 ^cdef^	11.60 ± 1.22 ^bcde^
	15	73.05 ± 3.69 ^bcde^	0.74 ± 0.06 ^bcdef^	12.19 ± 1.06 ^bc^
	20	72.63 ± 4.47 ^b^	0.80 ± 0.07 ^bcdef^	12.20 ± 1.63 ^bc^

Different letters attached to the values within the same column indicate the significantly different at *p* < 0.05 level.

**Table 2 foods-12-02573-t002:** Texture characteristics of raw and cooked shiitake mushrooms.

Cooking Methods	Cooking Time	Hardness (N)	Cohesivene	Gumminess (N)	Springiness (%)	Chewiness (N)	Adhesiveness (N.s)
Raw	0	2007.33 ± 10.97 ^a^	0.76 ± 0.06 ^abc^	1516.20 ± 8.28 ^a^	0.88 ± 0.05 ^ab^	1328.70 ± 67.64 ^a^	−1.23 ± 0.28 ^a^
Steaming	5	1594 ± 43.15 ^bc^	0.66 ± 0.03 ^fg^	1058.64 ± 28.66 ^cd^	0.73 ± 0.03 ^def^	826.90 ± 14.90 ^efg^	−12.73 ± 0.73 ^cd^
	10	1569.95 ± 219.01 ^bcd^	0.66 ± 0.03 ^fg^	1038.20 ± 144.83 ^cde^	0.77 ± 0.06 ^cdef^	808.72 ± 75.32 ^e^	−13.41 ± 0.23 ^cd^
	15	1026.47 ± 116.70 ^fg^	0.67 ± 0.05 ^fg^	686.11 ± 78.00 ^fgh^	0.73 ± 0 ^cde^	538.84 ± 37.31 ^hi^	−14.83 ± 0.46 ^cde^
	20	747.27 ± 24.05 ^g^	0.70 ± 0.03 ^def^	523.65 ± 16.85 ^h^	0.72 ± 0.02 ^abc^	423.16 ± 40.77 ^jk^	−15.88 ± 1.50 ^def^
Boiling	5	1748.98 ± 108.75 ^abc^	0.70 ± 0.05 ^cdef^	1228.29 ± 76.37 ^bc^	0.87 ± 0.03 ^ab^	1069.41 ± 37.83 ^c^	−8.07 ± 1.58 ^abc^
	10	1562.09 ± 151.00 ^bcd^	0.70 ± 0.05 ^def^	1092.11 ± 105.57 ^cd^	0.86 ± 0.04 ^ab^	933.86 ± 47.67 ^d^	−11.36 ± 3.0 ^bcd^
	15	1235.42 ± 126.99 ^def^	0.71 ± 0.03 ^cdef^	876.30 ± 90.07 ^def^	0.84 ± 0.07 ^abc^	734.07 ± 63.60 ^fg^	−21.79 ± 5.50 ^efg^
	20	1019.25 ± 226.12 ^fg^	0.67 ± 0.04 ^fg^	680.01 ± 150.86 ^fgh^	0.82 ± 0.06 ^bcd^	557.46 ± 38.49 ^h^	−11.01 ± 2.54 ^bcd^
Air frying	5	1210.65 ± 145.53 ^g^	0.77 ± 0.03 ^ab^	926.46 ± 111.37 ^def^	0.90 ± 0.03 ^a^	838.3062 ± 27.26 ^e^	−9.85 ± 0.62 ^bcd^
	10	1082.71 ± 135.43 ^efg^	0.73 ± 0.03 ^bcde^	792.98 ± 99.18 ^efg^	0.89 ± 0.05 ^a^	706.18 ± 37.00 ^g^	−15.77 ± 2.70 ^def^
	15	789.47 ± 260.08 ^g^	0.75 ± 0.04 ^abcd^	590.95 ± 194.68 ^gh^	0.90 ± 0.04 ^a^	529.21 ± 23.52 ^hi^	−23.11 ± 1.26 ^fg^
	20	751.44 ± 39.78 ^g^	0.62 ± 0.04 ^g^	467.58 ± 24.75 ^h^	0.80 ± 0.10 ^bcd^	374.53 ± 48.03 ^k^	−26.01 ± 5.91 ^g^
Oven baking	5	1803.00 ± 244.42 ^ab^	0.76 ± 0.03 ^a^	1442.90 ± 195.60 ^ab^	0.87 ± 0.06 ^ab^	1256.26 ± 89.33 ^b^	−4.57 ± 1.71 ^ab^
	10	1433.00 ± 81.00 ^cde^	0.76 ± 0.03 ^ab^	1095.53 ± 61.92 ^cd^	0.73 ± 0.08 ^def^	795.76 ± 92.76 ^ef^	−16.06 ± 2.46 ^def^
	15	913.29 ± 117.54 ^fg^	0.74 ± 0.05 ^bcde^	681.45 ± 75.95 ^fgh^	0.70 ± 0.04 ^ef^	479.48 ± 25.75 ^ij^	−26.17 ± 1.06 ^g^
	20	1012.62 ± 251.53 ^fg^	0.69 ± 0.06 ^ef^	702.16 ± 174.41 ^fgh^	0.70 ± 0.04 ^f^	489.51 ± 30.19 ^hij^	−112.341 ± 16.08 ^h^

Different letters attached to the values within the same column indicate the significantly different at *p* < 0.05 level.

**Table 3 foods-12-02573-t003:** Effects of different cooking methods and times on volatile flavor compounds of shiitake mushrooms.

	Rentation Time (min)	Compounds	Relative Contents (%)																
			Raw	Steaming (min)				Boiling (min)				Air Frying (min)				Oven Baking (min)			
				5	10	15	20	5	10	15	20	5	10	15	20	5	10	15	20
Alcohols	27.38	3-Octanol	Nd	0.60	Nd	Nd	Nd	0.12	0.32	Nd	0.27	Nd	Nd	Nd	Nd	Nd	Nd	Nd	0.67
	29.78	1-Octen-3-ol	41.32	6.36	2.26	1.68	1.01	4.50	21.88	15.32	0.67	2.24	3.65	4.39	1.20	0.65	0.25	0.28	0.32
	18.75	Eucalyptol	0.01	0.17	0.22	0.21	Nd	Nd	0.12	Nd	Nd	Nd	Nd	Nd	Nd	Nd	Nd	Nd	Nd
	30.26	1-Heptanol	0.13	Nd	Nd	Nd	Nd	Nd	Nd	Nd	Nd	Nd	Nd	Nd	Nd	Nd	Nd	Nd	Nd
	31.49	1-Hexanol, 2-ethyl-	0.04	0.36	0.55	0.51	0.41	Nd	0.29	0.79	0.27	Nd	Nd	Nd	Nd	0.43	Nd	0.31	Nd
	34.31	1-Octanol	2.54	Nd	0.38	0.27	Nd	Nd	0.26	Nd	Nd	0.37	0.35	Nd	0.12	Nd	Nd	0.15	Nd
	35.41	3-Octen-1-ol	0.01	Nd	Nd	Nd	Nd	Nd	Nd	Nd	Nd	Nd	Nd	Nd	Nd	Nd	Nd	Nd	Nd
	36.51	trans-2-Undecen-1-ol	Nd	Nd	Nd	Nd	Nd	Nd	Nd	Nd	Nd	0.11	Nd	Nd	Nd	Nd	Nd	Nd	Nd
	36.52	Cyclooctyl alcohol	Nd	Nd	Nd	Nd	Nd	Nd	Nd	Nd	Nd	Nd	Nd	0.20	Nd	Nd	Nd	Nd	Nd
	36.62	2-Octen-1-ol, (E)-	0.80	Nd	Nd	Nd	Nd	0.07	0.18	Nd	Nd	Nd	Nd	Nd	Nd	0.13	Nd	Nd	Nd
	36.63	trans-2-Undecen-1-ol	Nd	Nd	Nd	Nd	Nd	Nd	Nd	Nd	Nd	Nd	Nd	Nd	Nd	Nd	Nd	Nd	Nd
	36.66	2-Nonen-1-ol, (E)-	Nd	Nd	Nd	Nd	Nd	Nd	Nd	Nd	Nd	Nd	Nd	4.36	Nd	Nd	Nd	Nd	Nd
	38.35	1-Nonanol	Nd	Nd	Nd	Nd	Nd	Nd	Nd	Nd	Nd	0.27	0.22	Nd	Nd	Nd	Nd	Nd	Nd
	47.66	Phenylethyl Alcohol	Nd	Nd	Nd	Nd	Nd	Nd	Nd	Nd	Nd	0.05	Nd	0.07	Nd	Nd	Nd	Nd	Nd
	54.31	Cedrol	Nd	0.02	0.27	0.23	0.27	Nd	Nd	Nd	0.15	Nd	Nd	Nd	Nd	Nd	Nd	Nd	Nd
	Total		44.891	7.50	3.68	2.90	1.70	4.68	23.06	16.11	1.36	3.04	4.22	9.02	2.12	1.35	0.39	0.74	0.99
Aldehydes	37.95	Benzeneacetaldehyde	Nd	Nd	Nd	Nd	Nd	Nd	Nd	Nd	Nd	0.25	Nd	Nd	Nd	Nd	Nd	Nd	Nd
	13.44	Hexanal	Nd	1.57	4.69	8.63	3.19	0.22	1.19	3.44	1.74	Nd	Nd	0.40	0.40	1.30	0.90	0.90	1.01
	17.62	Heptanal	Nd	0.35	0.52	0.51	0.40		0.37	0.55	0.22	0.08	0.07	0.09	Nd	0.11	Nd	0.21	0.12
	22.36	Octanal	0.23	0.80	0.76	0.67	0.71	0.06	0.29	0.41	0.16	0.36	0.29	0.20	0.08	0.28	0.29	0.32	0.21
	24.28	2-Heptenal, (E)-	Nd	0.24	Nd	Nd	Nd	Nd	Nd	Nd	Nd	Nd	Nd	Nd	Nd	Nd	0.14	Nd	Nd
	27.11	Nonanal	Nd	0.24	2.40	1.79	2.61	2.39	1.21	1.70	0.90	2.31	1.70	1.37	Nd	1.63	Nd	1.61	0.94
	28.83	2-Octenal, (E)-	2.40	0.24	0.12	0.20	Nd	Nd	0.12	Nd	Nd	0.95	Nd	1.88	0.44	Nd	0.12	Nd	Nd
	28.87	2-Dodecenal	Nd	Nd	Nd	Nd	Nd	Nd	Nd	Nd	Nd	Nd	Nd	Nd	Nd	0.32	Nd	0.17	Nd
	31.68	Decanal	0.13	0.49	0.37	0.52	0.33	0.17	0.28	Nd	0.42	0.31	0.21	0.25	0.15	0.45	0.30	Nd	0.40
	32.92	Benzaldehyde	0.04	0.21	0.34	Nd	Nd	Nd	0.13	Nd	Nd	0.23	0.12	0.38	0.10	Nd	Nd	0.19	0.10
	33.34	2-Nonenal, (E)-	Nd	0.32	Nd	Nd	Nd	Nd	Nd	Nd	Nd	0.18	0.33	Nd	Nd	0.31	Nd	0.22	0.16
	35.64	2,4-Octadienal, (E,E)-	0.03	Nd	Nd	Nd	Nd	Nd	Nd	Nd	Nd	Nd	Nd	Nd	Nd	Nd	Nd	Nd	Nd
	43.70	2-Phenylpropenal	Nd	Nd	Nd	Nd	Nd	Nd	Nd	Nd	Nd	0.07	Nd	Nd	Nd	Nd	Nd	Nd	Nd
	Total		2.82	4.46	9.20	12.33	7.25	2.84	3.60	6.10	3.44	4.50	2.72	4.57	1.18	4.39	1.75	3.63	2.93
Ketones	20.86	3-Octanone	8.83	1.47	0.76	2.84	Nd	Nd	2.22	1.26	0.18	Nd	7.73	8.96	1.25	0.36	0.14	0.23	0.99
	23.11	1-Octen-3-one	6.81	Nd	Nd	Nd	Nd	Nd	Nd	Nd	Nd	1.38	Nd	3.08	1.10	Nd	Nd	Nd	Nd
	24.76	5-Hepten-2-one, 6-methyl-	Nd	0.12	Nd	Nd	Nd	Nd	Nd	Nd	0.28	Nd	Nd	Nd	Nd	Nd	Nd	Nd	Nd
	31.57	2-Decanone	Nd	Nd	Nd	Nd	Nd	Nd	Nd	Nd	Nd	Nd	Nd	Nd	Nd	Nd	Nd	0.40	0.29
	35.86	2-Undecanone	0.04	Nd	Nd	Nd	Nd	Nd	Nd	Nd	Nd	0.07	0.08	0.06	Nd	Nd	Nd	Nd	Nd
	38.08	Acetophenone	0.05	0.27	0.47	1.74	0.34	Nd	0.16	0.47	Nd	Nd	0.13	0.18	Nd	0.19	Nd	Nd	Nd
	Total		15.74	1.86	1.22	4.58	0.34	Nd	2.38	1.73	0.47	1.4424	7.94	12.29	2.35	0.55	0.14	0.63	1.29
Sulfur	13.12	Disulfide, dimethyl	1.69	Nd	Nd	Nd	Nd	Nd	Nd	Nd	Nd	4.74	1.75	0.88	0.42	Nd	Nd	Nd	Nd
	27.00	Thiourea	Nd	Nd	Nd	Nd	Nd	Nd	Nd	Nd	Nd	0.10	0.03	Nd	Nd	Nd	Nd	Nd	Nd
	27.14	Carbonochloridodithioic acid, methyl ester	0.23	Nd	Nd	Nd	Nd	Nd	Nd	Nd	Nd	Nd	Nd	Nd	Nd	Nd	Nd	Nd	Nd
	38.54	Disulfide, methyl (methylthio)methyl	Nd	Nd	Nd	Nd	Nd	Nd	Nd	Nd	Nd	Nd	Nd	0.18	Nd	Nd	Nd	Nd	Nd
	26.74	Dimethyl trisulfide	3.51	0.10	Nd	Nd	Nd	Nd	Nd	Nd	Nd	27.72	15.44	7.63	1.27	Nd	Nd	Nd	Nd
	Total		5.43	0.10	Nd	Nd	Nd	Nd	Nd	Nd	Nd	32.56	17.22	8.70	1.69	Nd	Nd	Nd	Nd
			Nd	Nd	Nd	Nd	Nd	Nd	Nd	Nd	Nd	Nd	Nd	Nd	Nd	Nd	Nd	Nd	Nd
Benzene	11.90	Toluene	Nd	Nd	Nd	Nd	Nd	0.12	Nd	Nd	Nd	Nd	Nd	Nd	Nd	Nd	Nd	Nd	Nd
	41.84	Oxime-, methoxy-phenyl-_	2.14	10.53	11.96	14.42	14.51	8.03	20.34	13.16	21.02	6.89	7.67	6.35	8.60	12.55	26.42	20.0	17.79
	53.54	Butylated Hydroxytoluene	Nd	Nd	Nd	Nd	Nd	Nd	Nd	Nd	Nd	Nd	Nd	Nd	Nd	0.08	Nd	Nd	Nd
	Total		2.14	10.53	11.96	14.42	14.51	8.15	20.34	13.16	21.02	6.89	7.67	6.35	8.60	12.63	26.42	20.70	17.79
Alkanes	25.07	Dodecane, 2,6,10-trimethyl-	Nd	Nd	Nd	Nd	Nd	0.06	Nd	Nd	Nd	Nd	Nd	Nd	Nd	Nd	Nd	Nd	Nd
	27.08	Tetradecane	Nd	Nd	Nd	Nd	Nd	0.10	Nd	Nd	Nd	Nd	Nd	Nd	Nd	Nd	1.23	Nd	Nd
	31.39	Pentadecane	0.03	Nd	Nd	Nd	Nd	Nd	0.14	Nd	Nd	Nd	Nd	Nd	Nd	Nd	Nd	Nd	Nd
	31.66	Pentadecane	0.04	Nd	Nd	Nd	Nd	Nd	Nd	Nd	Nd	Nd	Nd	Nd	Nd	Nd	Nd	Nd	Nd
	35.63	Hexadecane	0.06	Nd	0.21	Nd	Nd	Nd	Nd	Nd	Nd	0.06	0.07	Nd	Nd	0.10	Nd	Nd	Nd
	39.67	Heptadecane	Nd	Nd	Nd	Nd	Nd	Nd	Nd	Nd	Nd	Nd	Nd	Nd	Nd	Nd	Nd	Nd	Nd
	40.25	Octadecanal	0.04	Nd	Nd	Nd	Nd	Nd	Nd	Nd	Nd	Nd	Nd	Nd	Nd	Nd	Nd	Nd	Nd
	47.88	Octadecanal	0.05	Nd	Nd	Nd	Nd	Nd	Nd	Nd	Nd	Nd	Nd	Nd	Nd	Nd	Nd	Nd	Nd
	49.41	Cyclododecane	0.23	Nd	Nd	Nd	Nd	Nd	Nd	Nd	Nd	Nd	Nd	Nd	Nd	Nd	Nd	Nd	Nd
	Total		Nd	Nd	0.21	Nd	Nd	0.17	0.14	Nd	Nd	0.06	0.07	Nd	Nd	0.10	1.23	Nd	Nd
Alkenes	9.19	1,3-Octadiene	0.03	Nd	Nd	Nd	Nd	Nd	Nd	Nd	Nd	Nd	Nd	Nd	Nd	Nd	Nd	Nd	Nd
	9.27	Cyclooctene	0.03	Nd	Nd	Nd	Nd	Nd	Nd	Nd	Nd	Nd	Nd	Nd	Nd	Nd	Nd	Nd	Nd
	17.87	D-Limonene	Nd	Nd	Nd	Nd	Nd	0.12	Nd	Nd	Nd	Nd	Nd	Nd	Nd	Nd	Nd	Nd	Nd
	38.35	1-Decene	Nd	Nd	Nd	Nd	Nd	Nd	Nd	Nd	Nd	Nd	Nd	0.18	Nd	Nd	Nd	Nd	Nd
	Total		0.06	Nd	Nd	Nd	Nd	0.12	Nd	Nd	Nd	Nd	Nd	0.18	Nd	Nd	Nd	Nd	Nd
Others	19.58	Furan, 2-pentyl-	Nd	Nd	Nd	Nd	Nd	Nd	Nd	0.41	0.17	Nd	Nd	Nd	Nd	0.07	Nd	Nd	Nd
	34.30	Formic acid, octyl ester	Nd	0.30	Nd	Nd	0.36	0.10	Nd	Nd	Nd	Nd	Nd	0.25	Nd	0.14	Nd	Nd	Nd
	38.37	Nonyl chloroformate	Nd	Nd	Nd	Nd	Nd	Nd	Nd	Nd	Nd	Nd	Nd	Nd	Nd	Nd	Nd	Nd	Nd
	50.80	Phenol	Nd	Nd	Nd	Nd	Nd	Nd	Nd	Nd	Nd	Nd	Nd	Nd	Nd	Nd	Nd	Nd	0.17
	57.47	Phenol, 2,6-bis(1,1-dimethylethyl)-	Nd	Nd	Nd	Nd	Nd	Nd	Nd	Nd	Nd	Nd	Nd	Nd	Nd	Nd	Nd	Nd	Nd
	Total		Nd	0.46	0.40	Nd	Nd	0.18	Nd	Nd	Nd	0.09	Nd	Nd	Nd	Nd	Nd	Nd	0.09
Total			71.60	24.76	26.73	34.64	24.16	8.04	49.68	37.52	26.46	48.49	39.93	41.35	15.93	19.23	29.94	25.42	22.99

“Nd” means “not detectable” for the compound.

## Data Availability

Data are contained within the article.

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
