# Peer review of "Effect of Different Cooking Methods on the Bioactive Components, Color, Texture, Microstructure, and Volatiles of Shiitake Mushrooms"

_foods, 2023, doi:10.3390/foods12132573_

Round 1

Reviewer 1 Report

Thank you very much for your interesting research.

Some points must be carefully revised:

1. ABSTRACT. Line 18. I consider that ‘fresh’ should be replaced by ‘raw’ to avoid confusion (not only in the abstract but throughout the whole text).

2. INTRODUCTION. Line 35. β-glucans and chitins must be mentioned as relevant examples of dietary fibers/polysaccharides.

3. INTRODUCTION. Lines 37-38. Hypocholesterolemic activity must be mentioned.

4. INTRODUCTION. Line 44. Which specific compounds remained intact?

5. MATERIALS & METHODS. Please, describe for each determination if you use fresh or dried samples (it is described for polysaccharide/eritadenine content, but it is not described for phenolics determination). In the case that you used fresh (non-dried) samples, could you explain how you calculated contents (since some cook methods led to higher water contents than others, e.g. boiling > oven baking)? In the case that you used dried samples, please indicate which drying method you used (freeze-drying?).

6. MATERIALS & METHODS. Line 95. Please, add information about phenol sulfuric method, or at least cite a previous work where the method was briefly described.

7. RESULTS & DISCUSSION. Line 169. Have you considered to determine the presence of TPCs and other bioactives (polysaccharides) in the boiling water to confirm the leaching phenomenon?

8. RESULTS & DISCUSSION. Lines 198-209. Since eritadenine is a water-soluble molecule, maybe these losses should be highlighted for boiling. Although you compared your results with previously published works, you should include this study (
10.1039/c8fo01704b
) that reported similar results (boiling and frying were detrimental for eritadenine).

9. RESULTS & DISCUSSION. What about potential bioaccessibility/bioavailability of the bioactive compounds? Do you think the methods that you applied have an impact on these parameters?

10. CONCLUSIONS. What about future perspectives? Is further research encouraged? Bioaccesibility/bioavailability studies? Biological activities tests?

Author Response

Reviewer1:

Thank you very much for your interesting research.

Response: Thank you for your nice comments on our article. According to your suggestions, we have supplemented several data here and corrected several mistakes in our previous draft. The detailed point-by-point responses are listed below.

Some points must be carefully revised:

Point 1. ABSTRACT. Line 18. I consider that ‘fresh’ should be replaced by ‘raw’ to avoid confusion (not only in the abstract but throughout the whole text).

Response: Thank you for your suggestion. We have modified “fresh” to “raw” in whole draft.

Point 2. INTRODUCTION. Line 35. β-glucans and chitins must be mentioned as relevant examples of dietary fibers/polysaccharides.

Response: As you suggested, we have supplemented the description of β-glucans and chitins in the revised manuscript in Page 1 Lines 39 of the manuscript (clean version).

Point 3. INTRODUCTION. Lines 37-38. Hypocholesterolemic activity must be mentioned.

Response: Thank you for your suggestion. We have supplemented the description of hypocholesterolemic activity in the revised manuscript in Page 1 Lines 41 (clean version).

Point 4. INTRODUCTION. Line 44. Which specific compounds remained intact?

Response: According to the study reported by Miglio et al. (2008) in broccoli, steam-cooking had no influence on vitamin C content, but can increase the contents of polyphenols as well as the main glucosinolates (Ref. 16). Therefore, the specific compound remained intact was vitamin C, and we added this information in the revised manuscript.

Point 5. MATERIALS & METHODS. Please, describe for each determination if you use fresh or dried samples (it is described for polysaccharide/eritadenine content, but it is not described for phenolics determination). In the case that you used fresh (non-dried) samples, could you explain how you calculated contents (since some cook methods led to higher water contents than others, e.g. boiling > oven baking)? In the case that you used dried samples, please indicate which drying method you used (freeze-drying?).

Response: Thank you for your professional review work on our article. We feel sorry that we did not provide enough information. In the revised manuscript, we added the details for sample preparation as follow: “Half of the samples were freeze-dried for bioactive compounds analysis (total phenol, total flavonoid, polysaccharides and eritadenine)…...”

To avoid the effect of water content in cooked mushrooms on the contents of active components, the samples ground into powders after lyophilizing, and the active component contents were detected and expressed as milligram or gram per gram of mushroom powders’ dry weight (e.g. mg/g DW).  

Point 6. MATERIALS & METHODS. Line 95. Please, add information about phenol sulfuric method, or at least cite a previous work where the method was briefly described.

Response: Thank you for your reminder. We have supplemented relevant literature in the revised manuscript [Nielsen, S.S. (2010). Phenol-Sulfuric Acid Method for Total Carbohydrates. In: Nielsen, S.S. (eds) Food Analysis Laboratory Manual. Food Science Texts Series. Springer, Boston, MA., 10.1007/978-1-4419-1463-7 (Chapter 6), 47–53].

Point 7. RESULTS & DISCUSSION. Line 169. Have you considered to determine the presence of TPCs and other bioactives (polysaccharides) in the boiling water to confirm the leaching phenomenon?

Response: As you concerned,. in the pre-experiment, we determined the contents of total phenol, total flavonois, crude polysaccharides, and eritadenine in the water after boiling for 10 min. And the results showed that the contents respectively were 6 mg/mL, 2.5 mg/mL, 3 mg/mL g and 3.5mg/mL for total phenol, total flavonoid, crude polysaccharides, and eritadenine in the water. These results proved that the present of leaching phenomenon.  Because the data were not tested under all conditions (boiling for 5, 10, 15 and 20 min), and only the contents of the four components were analyzed under other cooking methods, the contents in water were not listed in the manuscript.

Point 8. RESULTS & DISCUSSION. Lines 198-209. Since eritadenine is a water-soluble molecule, maybe these losses should be highlighted for boiling. Although you compared your results with previously published works, you should include this study (10.1039/c8fo01704b) that reported similar results (boiling and frying were detrimental for eritadenine).

Response: Thank you for your suggestion. We have supplemented relevant discussions about the study (Ref. 32) in the revised manuscript in Page 6 Lines 232 – 238 (clean version).

Point 9. RESULTS & DISCUSSION. What about potential bioaccessibility/bioavailability of the bioactive compounds? Do you think the methods that you applied have an impact on these parameters?

Response: Thank you for your professional review . Many studies have proved the bioactivity of the active compounds from shiitake mushroom, and some researches have confirmed that cooking affects the content and bioactivity of active substances. Therefore, we believe that the cooking methods used in the study have impacts on the bioavailability of the bioactive substances. Moreover, we have validated the influences by in vitro experiments, and the in vivo assays are conducting currently.

Considering the research purpose of this study, the influences of cooking on the components’ biological activities were not included in this paper, but we believe that the relevant study results will be released soon.

Point 10. CONCLUSIONS. What about future perspectives? Is further research encouraged? Bioaccesibility/bioavailability studies? Biological activities tests?

Response: Thank you for your suggestion. We enriched the conclusions, and complemented the limitations of this study and what needs to be conducted in the future study as shown in Page 14 Line 386-391 of the manuscript (clean version).

Reviewer 2 Report

Dear Editor/Author,

My comments were sticky on attached pdf file

Regards,

 English language is fine.

Author Response

Reviewer 2:

Response: Special thanks to you for your good comments. We have studied comments carefully and have made correction which we hope meet with approval. All the changes were marked using “Track Changes” mode.

The responses to the comments and questions are listed below.

Point 1. the reference is not found, as well, Is TPC was determine by using aluminumchloride ???

Response: Sorry for our carelessness. We have supplemented the reference in the text (Ref. 21). The detection method for TPC was Folin-Ciocalteu reagent(Page 2 Lines 91)of the manuscript (clean version).

Point 2. L.edodes What is that????

Response: This is the abbreviation of the Latin name (Lentinula edodes) of shiitake mushroom. In the revised manuscript, all Latin names are indicated by “shiitake” or “shiitake mushroom”.

Point 3. Pleas provide enough information about the instrument which used for analysis, consider that for whole manuscript..

Response: Thank you for your suggestion. We have supplemented the description of instrument which used for analysis in the whole revised manuscript .

Point 4. model, manufacturer, specs and country

Response: As you suggested, we have supplemented the description of model, manufacturer, specs and country of scanning electron microscope in the revised manuscript in Page 4 Lines 151 of the manuscript (clean version).

Point 5. Change to capital letter.

Response: We are very sorry for our incorrect writing. The initials of first columns in Table 1, 2, and 3 are all changed to capital letters.

Point 6. In this table, you should write "Nd" as mean "not detectable" for the compound which is absent

Response: Thank you for your suggestion. We have explained the “"Nd" as mean "not detectable" for the compound” as an annotation of Table 3.